# Melatonin Delays Postharvest Senescence through Suppressing the Inhibition of BrERF2/BrERF109 on Flavonoid Biosynthesis in Flowering Chinese Cabbage

**DOI:** 10.3390/ijms24032933

**Published:** 2023-02-02

**Authors:** Lingqi Yue, Yunyan Kang, Min Zhong, Dengjin Kang, Puyan Zhao, Xirong Chai, Xian Yang

**Affiliations:** College of Horticulture, South China Agricultural University, Guangzhou 510642, China

**Keywords:** flowering Chinese cabbage, melatonin, flavonoid biosynthesis, postharvest senescence, ethylene response factors

## Abstract

Flowering Chinese cabbage is prone to withering, yellowing and deterioration after harvest. Melatonin plays a remarkable role in delaying leaf senescence and increasing flavonoid biosynthesis. However, the underlying molecular mechanisms of melatonin procrastinating postharvest senescence by regulating flavonoid biosynthesis remain largely unknown. In this study, melatonin could promote flavonoid accumulation and delay the postharvest senescence of flowering Chinese cabbage. Surprisingly, we observed that *BrFLS1* and *BrFLS3.2* were core contributors in flavonoid biosynthesis, and BrERF2 and BrERF109 were crucial ethylene response factors (ERFs) through the virus-induced gene silencing (VIGS) technique, which is involved in regulating the postharvest senescence under melatonin treatment. Furthermore, yeast one-hybrid (Y1H), dual luciferase (LUC), and β-glucuronidase (GUS) tissue staining experiments demonstrated that BrERF2/BrERF109 negatively regulated the transcripts of *BrFLS1* and *BrFLS3.2* by directly binding to their promoters, respectively. Silencing *BrERF2/BrERF109* significantly upregulated the transcripts of *BrFLS1* and *BrFLS3.2*, promoting flavonoid accumulation, and postponing the leaf senescence. Our results provided a new insight into the molecular regulatory network of melatonin delaying leaf senescence and initially ascertained that melatonin promoted flavonoid accumulation by suppressing the inhibition of BrERF2/BrERF109 on the transcripts of *BrFLS1* and *BrFLS3.2*, which led to delaying the leaf senescence of postharvest flowering Chinese cabbage.

## 1. Introduction

Plant senescence is a complex and highly coordinated process, which obviously affects the quality of crops [1,2]. Leaf senescence is influenced by a variety of internal and external factors. For example, ethylene causes disruption of the antioxidant system, leading to the onset and development of leaf senescence, and even to the leaf necrosis and plant death [3,4,5]. Ethylene transcription factors (ERFs), as one of the largest plant transcription factors, play powerful roles in mediating the plant senescence [6,7]. For example, RhERF3 is strongly induced by ethylene, which directly interacts with 9-cis-epoxycarotenoid dioxygenase (*RhNCED1*) and activates its transcriptional activity, eventually speeding up the aging process of rose [8]. In Arabidopsis, AtERF34 directly activates the expressions of salt stress-responsive genes *AtERD10* and *AtRD29A*, which negatively regulates the leaf senescence [9]. OsERF101 directly regulates the transcripts of chlorophyll degradation gene (*OsNAP*) and the core factor myelocytomatosis oncogenes (*OsMYC2*) of the jasmonic acid signaling pathway to promote the occurrence and progress of leaf senescence in rice [10]. Likewise, in tomato, SlERF.F5 involves in regulating jasmonic acid-induced leaf senescence by interacting with SlMYC2 [11]. However, whether and how ERFs are involved in leaf senescence of flowering Chinese cabbage (*Brassica campestris* L. ssp. *Chinensis* var. *utilis* Tsen et Lee) is still largely unclear.

Melatonin is widely found in plants and plays an essential role in delaying the aging of crops [12,13,14,15,16]. Melatonin biosynthesis begins with tryptophan, and at least six enzymes (Tryptophan decarboxylase, tryptamine 5-hydroxylase, Serotonin N-acetyltransferase, arylalkyl amine N-acetyltransferase, N-acetyl-serotonin methyltransferase, and hydroxyindole-O-methyltransferase) have been known to be involved in the biosynthesis [17]. The major genes encoding these enzymes of the entire melatonin biosynthetic pathway involved in delaying aging have been identified from various plants, including tryptophan decarboxylase, tryptophan hydroxylase, tryptamine 5-hydroxylase, serotonin N-acetyltransferase, N-acetylserotonin methyltransferase and caffeic acid O-methyltransferase [18]. Melatonin contributes to prolonging the shelf life of broccoli and Chinese cabbage, and maintains high nutrient content during late storage [19,20,21]. Melatonin also can reduce ethylene synthase activity, restrict ethylene production, resulting in postponing the senescence and maintaining the fruit quality of pear and mango [14,22]. Furthermore, melatonin slows down the fruit softening during later storage by decreasing the expressions of *MdERF109* in apples, and *AdERF4*, *AdERF74*, and *AdERF75* in kiwifruit [23]. These findings provide sufficient proof that melatonin plays pivotal roles in delaying senescence, and regulates the senescence process by affecting ethylene release and the expression of ERFs. It is well known that ERFs family contains multiple family members, however, it has not identified which members are the pivotal regulators in procrastinating the senescence of flowering Chinese cabbage mediated by melatonin.

Flavonoid, a class of secondary metabolites and phytonutrients widely existing in plants, plays important role in growth and development, antioxidation, and senescence of plants [24,25]. Previous studies have shown that flavonoid accumulation delays postharvest aging (broccoli) and prolongs storage time (tomato) [26,27]. The high level of flavonoid and the expressions of flavonoid synthesis-related genes are beneficial to delaying the leaf senescence in rice [28]. Flavonoids also cross-talk with melatonin to delay aging [29,30]. For example, melatonin delays the senescence by enhancing the flavonoid biosynthesis and activating the antioxidant capacity in kiwifruit [29]. Flavonol is the most abundant and extensive distribution of flavonoid, and directly affects the total amount of flavonoid. Flavonol synthase (FLS), as a key enzyme in flavonol biosynthesis, directly determines the flavonol content in plant [31,32,33]. However, the molecular regulation mechanism of the critical *FLSs* involved in hindering leaf senescence mediated by melatonin is still unclear.

Flowering Chinese cabbage is widely cultivated in South China [34]. It is very prone to senescence, wilt and decay after harvest, and only be stored for 2–3 days in the air at ambient temperature due to its higher water content [35]. Nowadays, only a limited number of studies have explored the postharvest senescence of flowering Chinese cabbage under melatonin treatment exhibiting that leaf senescence is associated with antioxidant system activities and abscisic acid biosynthesis [20,21]. The current studies show that delaying leaf senescence is accompanied by the flavonoid accumulation in leaves [28]. So far, the underlying molecular mechanisms of melatonin involved in retarding the postharvest senescence of flowering Chinese cabbage by mediating ERFs to regulate the flavonoid biosynthesis-related genes have not been explored. We aimed to identify which ERFs and flavonoid biosynthesis-related genes are the pivotal candidates in delaying the postharvest senescence of flowering Chinese cabbage mediated by melatonin.

## 2. Results

### 2.1. Melatonin Delaying Leaf Senescence was Associated with Chlorophyll Degradation, Ethylene Biosynthesis, and Flavonoid Accumulation

Flowering Chinese cabbage is a kind of leaf vegetable, and leaf yellowing is a typical characteristic of plant senescence. In this study, exogenous melatonin treatment significantly hindered the postharvest senescence of flowering Chinese cabbage. The leaves of the control progressively turned yellow after 10 days of storage and had no commercial value at 20 days of storage. In contrast, melatonin treatment kept the leaves greener during storage, and the leaves did not turn yellow until 20 days of storage (Figure 1A). Furthermore, the total chlorophyll content of melatonin treatment was 8.53, 2.74, 26.72, and 83.18% higher than that of the control at 5, 10, 15, and 20 days of storage, respectively (Figure 1B). The changing trend of chlorophyll a, and b content was consistent with that of the total chlorophyll content under melatonin treatment (Appendix A). Likewise, melatonin treatment decreased the transcripts of chlorophyll catabolic, and senescence-related genes (*BrNYC1*, *BrNOL*, *BrPAO*, *BrRCCR*, *BrHCAR, BrMCS, BrPPH1*, *BrPPH2*, *BrSGR1*, *BrSGR2*, *BrSGR3*, and *BrSAG15*) compared with the control (Appendix A).

Ethylene has been identified as a hormone capable of directly enhancing leaf senescence [4]. In this study, we observed that ethylene production increased significantly in the control group during storage, but melatonin treatment reduced the ethylene production by 14.61, 18.77, 11.62, and 10.24% at 5, 10, 15, and 20 days of storage, respectively (Figure 1C). As expected, melatonin significantly suppressed the expressions of ethylene biosynthesis-related genes such as *BrSAMS2.1*, *BrSAMS2.2, BrACS5*, *BrACS10, BrACO2*, and *BrACO5* compared with the control (Appendix A).

Flavonoid are essential for the anti-oxidant and anti-senescence process of plants [24]. To confirm whether melatonin can regulate the formation of flavonoid, the total flavonoid content in flowering Chinese cabbage during the 20 days of storage was analyzed. Our results displayed that exogenous melatonin treatment significantly affected the flavonoid biosynthesis during storage, especially in late storage. The total content of flavonoid in melatonin-treated leaves was higher than that in the control, as evident by the reduced loss of total flavonoid from 10 to 20 days of storage (Figure 1D).

The aforementioned results demonstrated that exogenous melatonin delaying the senescence of postharvest flowering Chinese cabbage was related to inhibiting chlorophyll degradation and ethylene biosynthesis, and promoting flavonoid accumulation.

### 2.2. BrFLS1 and BrFLS3.2, Key Genes for Flavonoid Biosynthesis, Played a Positive Regulatory Role in Melatonin-Delayed Leaf Senescence

Flavonol synthase (FLS) is a key enzyme in flavonoid biosynthesis. To further confirm the roles of flavonoid in leaf senescence under melatonin treatment, we analyzed the transcripts of nine flavonoid biosynthesis-related genes (*BrPAL3*, *BrC4H*, *Br4CL*, *BrFLS1*, *BrFLS2*, *BrFLS3.1*, *BrFLS3.2*, *BrFLS3.3*, and *BrFLS4*). Figure 2 demonstrated that the transcripts of *BrPAL3*, *BrC4H*, *Br4CL*, *BrFLS1*, *BrFLS2*, *BrFLS3.1*, *BrFLS3.2*, and *BrFLS4* were higher in melatonin-treated than that in control during 20 days storage, especially, the expressions of *BrFLS1* and *BrFLS3.2* were most significantly up-regulated. However, there was no significant difference in the transcript of *BrFLS3.3* between melatonin treatment and control. Moreover, the expression of *BrFLS1* reached a peak on day 10 of storage, and the expression of *BrFLS3.2* reached the highest level on day 10 or 20 of storage in melatonin treatment. These findings suggested that the flavonoid biosynthesis-related genes were involved in regulating leaf senescence, while *BrFLS1* and *BrFLS3.2* might be the key candidate genes to modulate flavonoid biosynthesis under melatonin treatment.

To further illustrate the regulating functions of the mentioned above nine *BrFLSs* on flavonoid biosynthesis in the leaf senescence, we created transgenic flowering Chinese cabbage lines of TRV-*BrFLSs* using VIGS technology. Six transient silencing transgenic lines (TRV-*BrFLS1*, TRV-*BrFLS2*, TRV-*BrFLS3.1*, TRV-*BrFLS3.2*, TRV-*BrFLS3.3*, and TRV-*BrFLS4*) were obtained for a further experiment, and the silencing efficiency was verified by qRT-PCR (Figure 3A). We observed that the leaf senescence of flowering Chinese cabbage was dramatically accelerated after silencing *BrFLSs* (Figure 3B, Appendix A). At 20 days after silencing, the base leaves of the TRV-*BrFLSs* plants began to turn yellow, and the TRV-*BrFLS1* and TRV-*BrFLS3.2* plants exhibited more serious symptoms compared with the TRV2 plants (Appendix A). At 30 days after silencing, most of the leaves in TRV-*BrFLSs* plants exhibited serious yellowing phenotype or even fall off, especially in TRV-*BrFLS1* and TRV-*BrFLS3.2* plants (Figure 3B and Appendix A). Moreover, the contents of chlorophyll, and flavonoids in the TRV-*BrFLSs* plants were lower than that in the TRV2 plants (Figure 3C,D and Appendix A), especially, those of the TRV-*BrFLS1* and TRV-*BrFLS3.2* plants were significantly decreased, and the transcription levels of chlorophyll degradation genes (*BrNYC1*, *BrHCAR*, *BrRCCR*, *BrCLH*, *BrMCS*, *BrPPH*, and *BrSGR2*) were significantly up-regulated in the TRV-*BrFLS1* and TRV-*BrFLS3.2* plants (Appendix A).

Furthermore, spraying melatonin to TRV-*BrFLSs* plants slowed down the leaf senescence (Figure 3B and Appendix A) and partially suppressed the expressions of chlorophyll degradation-related genes (*BrNYC1*, *BrHCAR*, *BrRCCR*, *BrCLH*, *BrMCS*, *BrPPH*, and *BrSGR2*) (Appendix A). In contrast, the chlorophyll content in TRV-*BrFLS1* and TRV-*BrFLS3.2* plants treated with melatonin significantly increased by 55.06 and 45.74%, respectively (Figure 3C), and the flavonoid content significantly increased by 15.25 and 16.46%, respectively (Figure 3D). The above results further demonstrated that *BrFLS1* and *BrFLS3.2* were the pivotal genes involved in delaying leaf senescence and enhancing flavonoid biosynthesis, while melatonin could procrastinate the postharvest senescence by up-regulating the expressions of *BrFLS1* and *BrFLS3.2* to promote the flavonoid biosynthesis.

### 2.3. BrERF2 and BrERF109, the Transcriptional Suppressors, Were Involved in Leaf Senescence and Repressed by Melatonin

Given that melatonin could limit ethylene release, we speculated that melatonin delay senescence may be associated with some key ERFs. To confirm this hypothesis, we isolated nine ERFs (Bra024954, Bra022115, Bra017495, Bra035792, Bra018864, Bra038261, Bra001784, Bra011529, and Bra017656) from the Chinese cabbage Chiifu genome database. Multi-sequence alignment of the transcription factors amino acids disclosed that each sequence contained a conserved AP2 domain (Appendix A), which could bind to the GCC-box (GCCGCC) or DRE/CRT element (CCGAC), which are the typical characteristics of ERFs proteins. Phylogenetic tree analysis was performed between ERFs identified in Arabidopsis and Bra024954, Bra022115, Bra017495, Bra035792, Bra018864, Bra038261, Bra001784, Bra011529, and Bra017656 in the Chinese cabbage genome database, they were named as BrERF2-like1, BrERF2-like2, BrERF2, BrERF3-like1, BrERF3-like2, BrERF3-like3, BrERF3-like4, BrERF109-like, and BrERF109, respectively (Figure 4A). In addition, we observed that the expressions of the nine *BrERFs* increased dramatically with the extension of storage time, and reached the peak at 20 days of storage. Nevertheless, melatonin could suppress the expressions of these genes compared to the control, especially, the *BrERF2* and *BrERF109* among the nine *BrERFs* were most significantly suppressed, which would be the focus of our extended investigation (Figure 4B). In contrast, the transcripts of *BrERF2-like2*, *BrERF3-like2*, and *BrERF3-like4* differed relatively little between the control and melatonin treatment (Figure 4B). Interestingly, we obtained similar results after spraying melatonin on pre-harvest plants showing that the expressions of the nine *BrERFs* was markedly inhibited. More importantly, the expressions of *BrERF2* and *BrERF109* were most significantly repressed, followed by *BrERF2-like1*, *BrERF2-like2*, and *BrERF109-like1*, while *BrERF3-like1*, *BrERF3-like2*, *BrERF3-like3*, and *BrERF3-like4* were almost unaffected by melatonin (Figure 4C and Appendix A). These results further suggested that BrERFs accelerated the leaf senescence process, while BrERF2 and BrERF109 may be the crucial candidates, which were suppressed by melatonin.

To further verify the transcriptional activity of BrERF2 and BrERF109, we performed Y2H and LUC experiments. The Y2H assay displayed that that yeast strains containing BD-BrERF2, BD-BrERF109, negative control, or positive control could grow normally in SD-Trp medium, but only positive control could grow in SD-Trp/-His medium (containing X-α-gal) and appeared blue, the rest of the yeast cells could not survive (Appendix A). These results implied that BrERF2 and BrERF109 had no transcriptional activation activity. Furthermore, the LUC assay exhibited that the transcriptional activities of BrERF2 and BrERF109 in tobacco cells were significantly lower than that of negative control (Figure 4D,E). These findings demonstrated that BrERF2 and BrERF109 were transcriptional inhibitors.

### 2.4. BrERF2 and BrERF109 Directly Bound to the Promoters of BrFLSs and Suppressed Their Expression

Provided that the expressions of *BrFLS1* and *BrFLS3.2* were increased, but the ethylene production and the transcripts of *BrERF2* and *BrERF109* were markedly suppressed under melatonin treatment. We further analyzed whether BrERF2 and BrERF109 could directly bind to the promoters of *BrFLS1* and *BrFLS3.2*. We observed that the promoter regions of *BrFLS1* and *BrFLS3.2* contained at least one of the DRE motifs, which were possible binding cis-elements of BrERFs (Figure 5A). The Y1H assay further demonstrated that BrERF2/BrERF109 could directly bind to their promoters of *BrFLS1* and *BrFLS3.2*, respectively (Figure 5B). Moreover, we investigated the regulations of BrERF2/BrERF109 on *BrFLS1* and *BrFLS3.2* promoters by GUS staining experiment and LUC assay in *N. benthamiana* leaves. GUS staining analysis revealed that when *Pro35S::BrERF2/BrERF109* was co-infiltrated with *ProBrFLS1*::GUS or *ProBrFLS3.2*::GUS, the blue phenomenon was significantly lighter in the tobacco than that in the controls (null, *ProBrFLS1*, and *ProBrFLS3.2*) (Figure 5C,D). Likewise, the relative LUC/REN ratio indicated that promoter activity of *BrFLS1* and *BrFLS3.2* was significantly weakened in the presence of BrERF2 or BrERF109 (Figure 5E,F). These results indicated that both BrERF2 and BrERF109 could inhibit the transcripts of *BrFLS1* and *BrFLS3.2* promoters.

### 2.5. Silencing BrERF2 and BrERF109 Promoted Flavonoid Biosynthesis and Delayed Aging

In order to further determine that inhibiting BrERF2 and BrERF109 could promote the biosynthesis of flavonoid and delay leaf aging, we generated the TRV-*BrERF2* and TRV-*BrERF109* transgenic plants of flowering Chinese cabbage. The two transient silencing transgenic lines and control plants (TRV2) of flowering Chinese cabbage were obtained for subsequent investigations (Figure 6A). As expected, the expressions of *BrFLS1* and *BrFLS3.2* in the TRV-*BrERF2* and TRV-*BrERF109* plants were dramatically higher than that in the TRV2 plants (control) as detected by qRT-PCR (Figure 6B). Additionally, as shown in Figure 6C, the leaves of the TRV-*BrERF2* and TRV-*BrERF109* plants were greener than that of the control plants (TRV2). The content of chlorophyll in the transgenic plants was significantly higher than that in the control, and similar results were observed in the determination of flavonoid content (Figure 6D,E). These findings attested that BrERF2 and BrERF109 were involved in promoting the leaf senescence of flowering Chinese cabbage, and inhibiting the expressions of *BrERF2* and *BrERF109* could promote the transcriptions of *BrFLS1* and *BrFLS3.2* and the accumulation of flavonoid, thus delaying leaf aging. Collectively, our results further testified that melatonin could enhance the flavonoid accumulation by suppressing the inhibition of BrERF2 and BrERF109 on the transcription of *BrFLS1* and *BrFLS3.2*, which resulted in delaying the leaf senescence of flowering Chinese cabbage.

### 2.6. BrERF2 Interacted with BrERF109, Which Formed a BrERF2-BrERF109 Complex

As we have known, both BrERF2 and BrERF109 could inhibit the expression of target genes *BrFLS1* and *BrFLS3.2*, respectively, but it was not clear that whether they could interact with each other. To test this hypothesis, we firstly performed Y2H screen to investigate whether BrERF2 could interact with BrERF109. Interestingly, the results showed that they could physically interact with each other in yeast cells (Figure 7A). To further verify this interaction in plant cells, we observed that YFP fluorescence was only detected in the nucleus of tobacco leaf cells harboring BrERF2-nYFP and BrERF109-cYFP by bimolecular fluorescence complementation (BiFC) experiments, but not in cells transformed with BrERF2-nYFP and cYFP, BrERF109-cYFP and nYFP or nYFP and cYFP (Figure 7B). Moreover, subcellular localization analysis indicated that BrERF2 and BrERF109 were consistently localized in the nucleus (Figure 7C). The aforementioned results suggested that BrERF2 and BrERF109 are nucleus-localized transcriptional suppressors. Unfortunately, although BrERF2 physically interacted with BrERF109, which formed a BrERF2-BrERF109 complex, they did not synergically inhibit the expressions of *BrFLS1* and *BrFLS3.2* (Appendix A).

## 3. Discussion

Flowering Chinese cabbage, which belongs to Brassica, is one of the largest cultivated leaf vegetables in China. However, because its edible parts are tender stems and young leaves, it is easy to lose water, wilt, yellow, and cause nutritional degradation after harvest, which seriously affects its postharvest quality [34,36]. Previous studies have indicated that plant senescence is associated with chlorophyll degradation and ethylene production, while melatonin can effectively procrastinate the senescence process in postharvest flowering Chinese cabbage through delaying chlorophyll breakdown, reducing expressions of senescence-related genes, and enhancing the antioxidant system [20,21,37,38,39]. Other reports also showed melatonin delay the aging of pear fruit by repressing the expressions of ethylene biosynthesis-related genes *PcACS1* and *PcACO1* [29]. These findings were similar to our results exhibiting that melatonin could effectively attenuate the leaf yellowing of postharvest flowering Chinese cabbage, which by reducing the chlorophyll degradation, and suppressing the expressions of chlorophyll degradation-related genes and senescence-associated gene (*BrNYC1*, *BrNOL*, *BrPAO*, *BrRCCR*, *BrHCAR, BrMCS, BrPPH1*, *BrPPH2*, *BrSGR1*, *BrSGR2*, *BrSGR3*, and *BrSAG15*) (Figure 1A,B, Appendix A). These results further confirmed that melatonin was a potent anti-senescent compound that could be employed to extend the shelf-life and maintain the quality of postharvest horticultural products. Nevertheless, to date, the molecular regulatory network of melatonin delaying the senescence of flowering Chinese cabbage remains predominantly insufficient.

Flavonoid is an important active phytochemical and antioxidant that is widely found in plants benefiting to human health [25,36,40]. Our recent studies demonstrated that the flavonoid biosynthesis was closely associated with the senescence of postharvest flowering Chinese cabbage [34]. It was reported that melatonin can promote flavonoid biosynthesis during the leaf senescence of kiwifruit [29]. So far, although melatonin can regulate the flavonoid content and the expressions of flavonoid biosynthesis-related genes in the aging process, it is not clear which genes play key roles and in what way they regulate leaf aging [29]. Here, we found that melatonin not only increased flavonoid accumulation but also up-regulated the expressions of flavonoid biosynthesis-related genes (*BrPAL3*, *BrC4H*, *Br4CL*, *BrFLS1, BrFLS2*, *BrFLS3.1*, *BrFLS3.2*, and *BrFLS4*) during postharvest senescence, especially the *BrFLS1* and *BrFLS3.2* (Figure 1D and Figure 2). Thus, we speculated that *BrFLS1* and *BrFLS3.2* were key flavonoid biosynthesis genes and potential regulators in delaying the leaf senescence mediated by melatonin. To verify whether *BrFLS1* and *BrFLS3.2* were involved in the regulation of leaf senescence, the VIGS system was performed in flowering Chinese cabbage. We observed that silencing *BrFLS1* and *BrFLS3.2* resulted in severe leaf yellowing, a dramatic decrease of flavonoid and chlorophyll content, and increasing the transcripts of chlorophyll degradation genes and senescence marker gene (*BrNYC1*, *BrNOL*, *BrPAO*, *BrRCCR*, *BrHCAR, BrMCS, BrPPH1*, *BrPPH2*, *BrSGR1*, *BrSGR2*, *BrSGR3*, and *BrSAG15*). Interestingly, melatonin reversed the onset of this phenomenon in *BrFLS1*-silenced and *BrFLS3.2*-silenced plants, thus delaying the leaf senescence (Figure 3 and Appendix A). In contrast, the *BrFLS2*, *BrFLS3.1*, *BrFLS3.3*, and *BrFLS4* did not play key roles in promoting flavonoid biosynthesis and delaying the leaf senescence mediated by melatonin, and may be present functional redundancy (Appendix A). Taken together, *BrFLSs* were involved in delaying the leaf senescence, and melatonin could enhance the flavonoid accumulation to delay the leaf senescence by up-regulating the expressions of *BrFLSs*, while the *BrFLS1* and *BrFLS3.2* were core contributors in flavonoid biosynthesis pathway. Nonetheless, the upstream transcription factors of *BrFLS1* and *BrFLS3.2* and their regulation mechanism associated with senescence remain undisclosed.

Ethylene as a negative regulatory signal factor, plays a vital regulatory role in the complex senescence process [4,5], while melatonin antagonizes ethylene to delay the senescence process in horticultural crops [14]. Zhai et al. [22] showed that melatonin delays the senescence of pear fruit by repressing the ethylene production and the expressions of ethylene biosynthesis genes *PcACS1* and *PcACO1*. Our present study had similar findings showing that melatonin could significantly inhibit the ethylene release and the expressions of ethylene biosynthesis-related genes (*BrSAMS2.1*, *BrSAMS2.2, BrACS5*, *BrACS10, BrACO2*, and *BrACO5*) (Figure 1C and Appendix A). ERFs are ethylene-responsive transcription factors, which regulate a variety of plant-specific processes. Nowadays, the intention of ERFs has been raising in regulating secondary plant metabolism and plant senescence [7]. Recently, ERF2 plays important roles in seed germination, regulation of calcium deficiency in postharvest fruits, and biotic stresses [13,41,42]. ERF109 regulates lateral root formation, anthocyanin synthesis, and mediates plant responses to abiotic stresses [43,44,45]. Nevertheless, the roles of ERF2 and ERF109 have not yet been explored in the postharvest senescence. In the present study, we isolated nine BrERFs members (BrERF2-like1, BrERF2-like2, BrERF2, BrERF3-like1, BrERF3-like2, BrERF3-like3, BrERF3-like4, BrERF109-like, and BrERF109) of flowering Chinese cabbage (Figure 4B), and found that the BrERFs negatively regulated leaf senescence, however, melatonin could antagonize their transcriptions, especially the BrERF2 and BrERF109, implying that they were key BrERFs involved in modulating the senescence of postharvest flowering Chinese cabbage under melatonin treatment.

Given that down-regulating *BrERF2* and *BrERF109*, and up-regulating *BrFLS1* and *BrFLS3.2* could delay the leaf senescence of postharvest flowering Chinese cabbage, however, available information about their interactions to regulate aging is still limited. To verify whether BrERF2 and BrERF109 were involved in the regulation of *BrFLS1* and *BrFLS3.2* to delay the leaf senescence, the combination experiments of Y1H, GUS straining assays, and LUC experiences were executed. In this work, we observed that BrERF2 and BrERF109 directly bound to the promoters of *BrFLS1* and *BrFLS3.2* in vitro and in vivo, respectively, and inhibited their expression (Figure 5). However, these inhibitory effects were significantly attenuated by melatonin, which implied that melatonin could inhibit BrERF2/BrERF109 mediated flavonoid biosynthesis (Figure 4B,C). Moreover, we were pleased to discover that *BrFLS1* and *BrFLS3.2* were significantly up-regulated in the *BrERF2*-silenced and *BrERF109*-silenced plants, accompanied by a significant increase in chlorophyll and flavonoid contents, maintaining greener leaves in the silenced plants (Figure 6). Taken together, these findings further confirmed that melatonin could promote the flavonoid accumulation by suppressing the inhibition of BrERF2/BrERF109 on the expressions of *BrFLS1* and *BrFLS3.2*, thus delaying the aging of postharvest flowering Chinese cabbage.

Transcriptional regulation is a dynamic network; numerous previous studies found that ERFs regulate the transcription of target genes by binding to their cis-acting elements [6]. A previous study showed that MdERF2 inhibits *MdACS1* transcription, and directly interacts with MdERF3 to suppress the binding of MdERF3 with *MdACS1* promoter, thus negatively affecting ethylene biosynthesis and fruit ripening in apples [46,47]. In soybean, GmDREB1 forms heterodimers with ERFs to regulate plant drought resistance [48]. In this study, we observed that BrERF2 physically interacted with BrERF109 and formed a complex (Figure 7A,B), however, they did not further synergically inhibit the expressions of *BrFLS1* and *BrFLS3.2* (Appendix A). Therefore, whether BrERF2 and BrERF109 have synergistic regulations of delaying aging through mediating flavonoid biosynthesis needs to be further explored.

## 4. Materials and Methods

### 4.1. Plant Materials and Treatments

The ‘60 days’ variety of flowering Chinese cabbage were used in this study. The fresh samples were harvested from Huachuang Farm (Zhongshan, China). To determine the optimal concentration of melatonin that can delay the leaf senescence of postharvest flowering Chinese cabbage, a pre-experiment was conducted. It was demonstrated that 100 μmol L^−1^ of melatonin (Sigma-Aldrich, Saint Louis, MO, USA) was a more effective dose to postpone the leaf senescence of postharvest Chinese flowering cabbage than treating with 0, 50, and 200 μmol L^−1^ (Appendix A). Accordingly, 100 μmol L^−1^ of melatonin was selected for the subsequent experiments. The experiments were divided into two groups. One group was sprayed with 100 μmol L^−1^ melatonin and another group was sprayed with distilled water (control). After treatment, the samples were put into a polyethylene bags (40 cm × 30 cm) and stored promptly at 4 °C. Each treatment group contained three biological replicates and 30 plants per replicate. Relevant physiological and molecular indexes were investigated at 0, 5, 10, 15, and 20 days of storage, respectively.

Tobacco was grown in an artificial light incubator of 23 °C (16 h light/8 h dark), and five-week-old plants were selected for *Agrobacterium tumefaciens*-mediated transient expression experiments.

### 4.2. Determination of Ethylene Content

Ethylene production of flowering Chinese cabbages during storage was measured according to the method of Zhou et al. [49]. In brief, ten plants of flowering Chinese cabbage were placed in a 10 L closed container with a gas sampling port, and the ethylene gas was collected with a syringe (1 mL). Then, the ethylene samples were infused into the headspace vial and the ethylene concentration was determined using a gas chromatograph (CP-3800, Varian, Palo Alto, CA, USA) equipped with a flame ionization detector (FID) and fitted with the chromatographic column (GDX-102, 3 m × 2 mm i.d., Dalian Institute of Chemical Physics, Chinese Academy of Sciences, Dalian, China). Chromatographic conditions were as follows: column temperature 85 °C; injector temperature 120 °C; detector temperature 200 °C; carrier gas nitrogen (N_2_), N_2_, hydrogen and compressed air at flow rates of 20, 30, and 300 mL min^−1^, respectively. The sample was injected at a flow rate of 1.0 mL min^−1^ and the determination was repeated three times. The concentration of ethylene was quantified using an external standard method and results were expressed as μmol kg^−1^ h^−1^.

### 4.3. Determination of Chlorophyll and Total Flavonoid Content

Chlorophyll a, chlorophyll b, and total chlorophyll contents of flowering Chinese cabbage were performed by previously reported methods [50]. Briefly, 0.5 g of fresh leaves were immersed in a mixed solution of acetone and ethanol (*v*:*v* = 1:1) until the leaves turned white (after 24 h). Then, the values of OD_645_ and OD_663_ of the extract were calculated by spectrophotometer (Uv-1800, Shimadzu, Shanghai, China).

The total flavonoid content of flowering Chinese cabbage was determined according to the previous method and modified moderately [51]. In brief, 0.5 g of samples were ground into powder in liquid nitrogen, then the powder was transferred to 8 mL absolute ethanol and stood for 30 min. The extract was centrifuged at 10,000 g for 15 min (4 °C). Rutin was used as the standard substance. The supernatant was gathered for the determination of the total flavonoid content. The samples were measured for total flavonoid content at a wavelength of 510 nm using UV-2410PC spectrophotometer (Shimadzu Corporation, Kyoto, Japan).

### 4.4. Total RNA Extraction, Quantitative Real-Time PCR (qRT-PCR) and Sequence Analysis

Total RNA was isolated from the flowering Chinese cabbage leaves using a TRIzol kit (ComWin Biotech Co., Ltd., Beijing, China) following the manufacturer’s protocol. The first-strand cDNA was synthesized from 1 μg of total RNA, after digestion of genomic DNA, using PrimeScript™ RT Master Mix (Takara Biomedical Technology Co. Ltd, Dalian, China). Then, cDNA was used as a template for quantitative real-time PCR (qRT-PCR) analysis. qPCR was carried out with the Q711-ChamQTM Universal SYBR^®^ qPCR Master Mix (Vazyme, Nanjing, China) on Bio-Rad CFX96 Real-Time PCR System. qPCR was executed using 0.4 μL (10 μM) of forward and reverse primers, 10 μL 2 × ChamQ Universal SYBR qPCR Master Mix and 2 μL of template cDNA in a total volume of 20 μL. qPCR reaction conditions were described as follows: 30 s at 95 °C, 40 cycles of 10 s at 95 °C and 30 s at 60 °C. The actin used as a reference gene, gene-specific primers designed according to cDNA sequences as shown in Appendix A. Relative gene expression levels were computed as described by previous [52], three different biological samples and three technical replicates were used for all quantitative experiments.

The Chinese cabbage Chiifu genome database (http://brassicadb.org/brad/, accessed on 16 September 2022) was used in this study. Sequence analysis was performed by utilizing MEGA7.0 and GeneDoc software to construct phylogenetic trees and multiple alignments, respectively.

### 4.5. Virus-Induced Gene Silencing (VIGS)

Fragments (300 bp) of BrFLSs (BrFLS1, BrFLS2, BrFLS3.1, BrFLS3.2, BrFLS3.3, and BrFLS4) and BrERFs (BrERF2-like1, BrERF2-like2, BrERF2, BrERF3-like1, BrERF3-like2, BrERF3-like3, BrERF3-like4, BrERF109-like, and BrERF109) were predicted by way of the VIGS tool on Solanaceae Genomics Network (https://solgenomics.net/, accessed on 9 October 2022) and amplified by PCR. The fragments were constructed on the TRV2 vector, then the recombinant plasmid (TRV2::BrFLSs, TRV2::BrERFs) and TRV2 (empty vector) were converted into Agrobacterium GV3101, respectively. The detailed experimental procedure is as described previously [53]. The injected plants were cultured at 23 ± 2 °C in an artificial light source incubator for 3 days under darkness and then switched to normal photoperiod culture (16 h light/8 h dark). Periodically, qPCR testing of flowering Chinese cabbage leaves was carried out, and silent plants were selected for further analysis.

### 4.6. Subcellular Localization Analysis

The coding sequences (CDS) of *BrERF2* and *BrERF109* full-length without the stop codon were amplified and fused with the pCAMBIA1300 vector. The recombinant plasmids (35S:BrERF2-GFP and 35S:BrERF109-GFP) or the control vector (35S: GFP), and nuclear marker NLS-RFP (NLS, nuclear localization sequence) were co-transformed into Agrobacterium. Then, the Agrobacterium containing 35S:BrERF2-GFP, 35S:BrERF109-GFP or 35S: GFP was injected into the suction of 4- to 6-week-old tobacco (*Nicotiana benthamiana*) blade with a 1 mL syringe without needle. The GFP fluorescence was observed by Axioskop 2 Plus fluorescence microscope (Zeiss, Oberkochen, Germany) at 48 h of infiltration [20].

### 4.7. Transcriptional Assay

The full-length coding sequences of *BrERF2* and *BrERF109* were fused with GAL4 DNA-binding domain into pGBKT7 vector. Then, the recombinant pGBKT7-BrERF2 and pGBKT7-BrERF109 fusion plasmids, positive control (pGBKT7-53 + pGADT7-T), or negative control (pGBKT7) were transformed into yeast two-hybrid (Y2H) Gold strain using the Yeastmaker Yeast Transformation System 2 (Takara Biomedical Technology Co., Ltd., Dalian, China), respectively. The trans-activation activity of BrERF2 and BrERF109 were assayed on basis of the growth status and α-galactosidase activity of yeast cells grown on SD/-Trp media or SD/-Trp-His-Ade media. The experiment was performed in triplicate.

The full-length regions of *BrERF2* and *BrERF109* were individually fused to the GAL4 DNA-binding domain of the pBD vector and acted as effector proteins. The reporter vector was modified from the pGreenII 0800-LUC vector, which included firefly luciferase (LUC) driven by a 35S promoter with five repeats of the upstream activating sequence, and a 35S promoter internal control of sub-driven renilla luciferase (REN). Each pair of effector and reporter plasmids was transiently transformed into tobacco leaves by Agrobacterium infiltration. After 48 h, the activities of LUC and REN were evaluated in Luminoskan Ascent microplate luminometer (Thermo Fisher Scientific, Waltham, MA, USA) using a LUC assay kit (Promega Corporation, Fitchburg, WI, USA). Eventually, the transcriptional activity of BrERF2 and BrERF109 was reflected by the value of LUC/REN. Six replicates were used for each experiment.

### 4.8. Yeast One-Hybrid (Y1H) Assay

The coding regions of *BrERF2* and *BrERF109* were cloned as bait constructs into the pGADT7 vector. The *BrFLS1* (1125 bp upstream of the predicted translation start site) and *BrFLS3.2* (570 bp upstream of the predicted translation start site) promoter fragments were ligated into the pAbAi vector as prey constructs. First, the bait constructs of pAbAi-*BrFLS1* and pAbAi-*BrFLS3.2* were transformed into Y1H yeast cells and cultured on SD/-Ura medium. Second, these preys and baits were co-transformed into Y1H yeast cells and cultivated on the SD/-Leu (Aba) medium using the Yeastmaker Yeast Transformation System 2 according to the manufacturer’s protocol.

### 4.9. β-Glucuronidase (GUS) Analysis

The CDS segments of *BrERF2* and *BrERF109* were cloned into the pRI101 vector utilizing specific primers to generate the effector construct. The *BrFLS1* (1125 bp) and *BrFLS3.2* (570 bp) promoter fragments were cloned into the pBI101 vector containing the 35S promoter as reporter constructs. The reporters and effectors were transformed into Agrobacterium strain GV3101, and the bacterial liquid was co-infiltrated into N. benthamiana. The GUS staining experiment was performed with reference to Ma et al. [45]. Permeabilization for each experiment was executed in three biological replicates.

### 4.10. Dual Luciferase (LUC) Assay

The full-length CDS sequences of *BrERF2* and *BrERF109* were inserted into pGreenII 62-SK vector as effectors, and the promoter fragments of *BrFLS1* and *BrFLS3.2* were inserted into pGreenII 0800-LUC vector as reporters. These recombinant plasmids were transformed into Agrobacterium, and the Agrobacterium solution containing effectors and reporters were infected into tobacco leaves. The empty vector PGreenII 62-SK was used as a negative control. LUC activity was measured using the LUC gene test kit (Promega). Six biological replicates were conducted.

### 4.11. Y2H Screening

The coding regions of *BrERF2* and *BrERF109* were amplified and cloned into pGADT7 and pGBKT7 vectors, respectively. The fusion plasmid was co-transformed into strain Y2H Gold according to the rules of Yeastmaker Yeast Transformation System 2 and grown on DDO (SD/-Leu/-Trp) medium for 3 days. The transformed colonies were spotted on the selective medium QDO (SD/-Leu/-Trp/-Ade/-His) containing AbA and X-α-gal, to assess the possible interactions according to their growth status and colony turning blue.

### 4.12. Bimolecular Fluorescence Complementation (BiFC) Assay

The CDS of *BrERF2* was integrated into the p2YC vector using the BamHI and SalI sites. The CDS of *BrERF109* was linked into the p2YN vector using the BamHI and SalI sites [54]. These recombinant plasmids were injected into Agrobacterium, then the bacterial fluid infiltrated into the N. benthamiana leaves. The injected tobacco was incubated in the dark for 48 h, and YFP fluorescence was observed with a fluorescence microscope (Zeiss Axioskop 2 plus). This instantaneous expression experiment was repeated at least three times and shown representative results.

### 4.13. Statistical Analysis

All data were expressed as mean ± SD. R v4.1.1 and IBM SPSS Statistics 24 (SPSS Inc., Chicago, IL, USA) were used for statistical analysis. A Tukey’s test (*p* ≤ 0.05) was performed to evaluate the treatment effects. All figures were generated using R v4.1.1 and Origin 2021 (OriginLab Corp., Northampton, MA, USA).

### 4.14. Primers

Details of the primers involved in Appendix A.

## 5. Conclusions

In this study, a new regulatory pathway of melatonin delaying postharvest senescence was observed. We found that flavonoid could delay the plant senescence of flowering Chinese cabbage. Suppressing BrERF2 and BrERF109, and enhancing *BrFLS1* and *BrFLS3.2* expressions could promote flavonoid biosynthesis. BrERF2 and BrERF109 directly targeted the promoters of *BrFLS1* or *BrFLS3.2* and restrained their transcripts. Melatonin could inhibit the transcripts of *BrERF2/BrERF109* and attenuate the inhibitory effects of ERFs on the transcripts of *FLSs*, ultimately increasing the level of flavonoid. Taken together, our results provided a novel insight into the molecular regulatory mechanism of melatonin involved in delaying postharvest senescence and confirmed for the first time that melatonin could enhance flavonoid biosynthesis by suppressing the inhibition of BrERF2/BrERF109 on the transcript of *BrFLS1* and *BrFLS3.2*, which resulted in promoting flavonoid accumulation and delaying the postharvest senescence of flowering Chinese cabbage. Our findings enrich the molecular regulatory network of melatonin involved in delaying plant senescence, and might have important practical implications for improving commercial product qualities and greatly assist in extending the shelf life of postharvest crops.

## Figures and Tables

**Figure 1 ijms-24-02933-f001:**
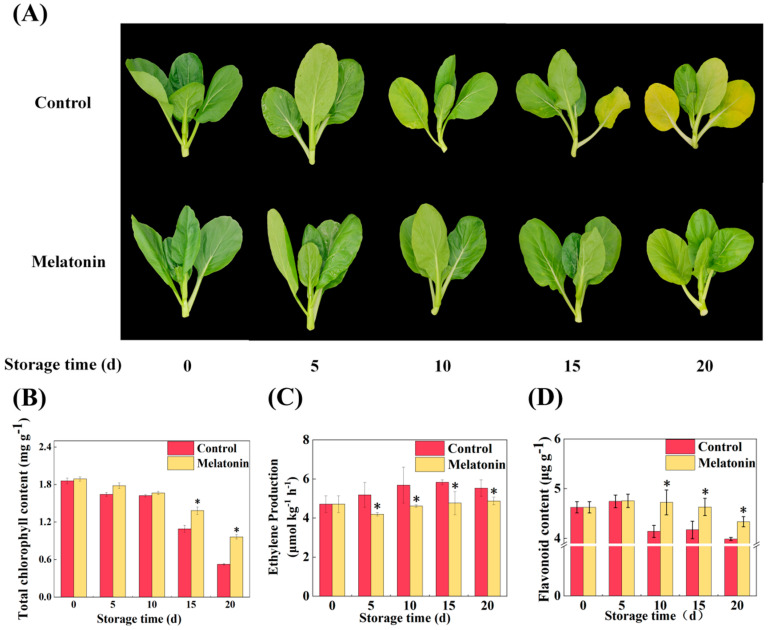
The effects of exogenous melatonin on the phenotype, chlorophyll content, ethylene production and flavonoid content of flowering Chinese cabbage at 4 °C during 20 days storage. (**A**) The phenotype, (**B**) total chlorophyll content, (**C**) ethylene production, (**D**) flavonoid content. Control: 0 μmol L^−1^ melatonin (pure water). Melatonin: 100 μmol L^−1^ melatonin treatment. The vertical bars indicate the standard error of the mean (n = 3). The asterisks above columns indicate a significant difference at * *p* < 0.05.

**Figure 2 ijms-24-02933-f002:**
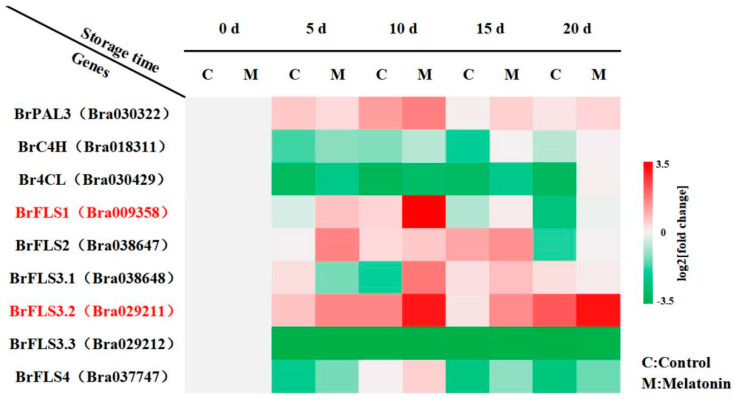
The effects of exogenous melatonin on flavonoid synthesis gene expression in flowering Chinese cabbage at 4 °C during 20 days storage. The heatmap indicates the log2-(fold change) levels, red: increase; green: decrease.

**Figure 3 ijms-24-02933-f003:**
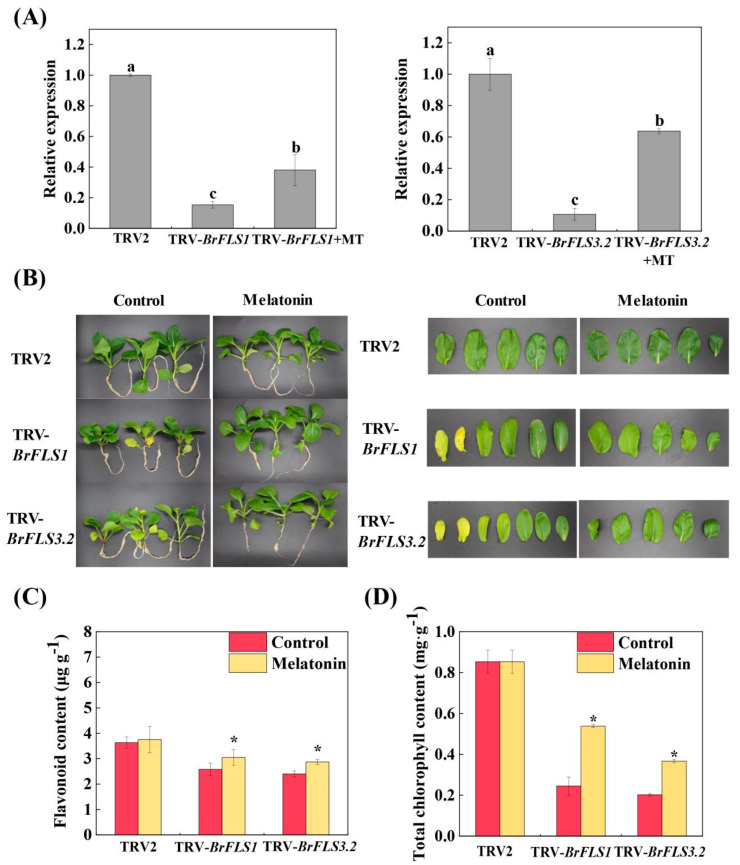
VIGS assay of *BrFLS1* and *BrFLS3.2* in flowering Chinese cabbage. (**A**), The relative transcript levels of *BrFLS1* and *BrFLS3.2* in TRV-*BrFLS1* and TRV-*BrFLS3.2* plants. (**B**), The plant and leaf phenotypes of TRV2, TRV-*BrFLS1* and TRV-*BrFLS3.2* plants. (**C**), Flavonoid content of TRV2, TRV-*BrFLS1* and TRV-*BrFLS3.2* plants. (**D**), Total chlorophyll content of TRV2, TRV-*BrFLS1* and TRV-*BrFLS3.2* plants. Control: 0 μmol L^−1^ melatonin (pure water). Melatonin: 100 μmol L^−1^ melatonin treatment. The vertical bars indicate the standard error of the mean (n = 3). The letters or asterisks above columns indicate a significant difference at * *p* < 0.05.

**Figure 4 ijms-24-02933-f004:**
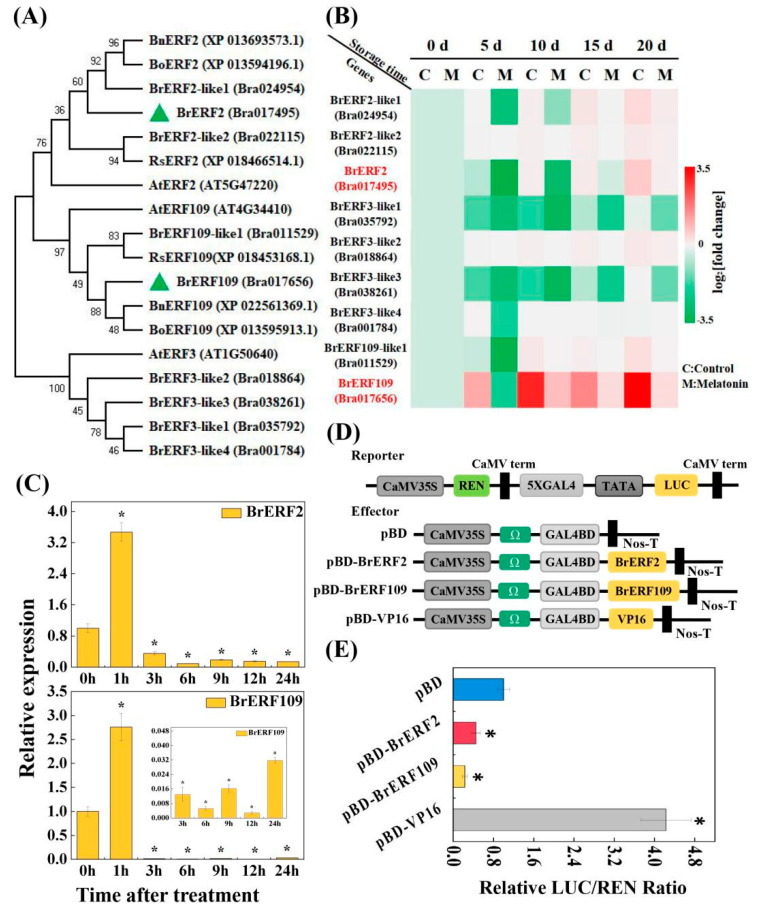
Identification and expression pattern analysis of *BrERFs*. (**A**), Phylogenetic tree analysis between ERFs protein in published species and BrERFs protein of flowering Chinese cabbage. The phylogenetic analysis was performed using the MEGA7.0 software. (**B**), The differential expression levels of nine *BrERFs* under melatonin treatment. The heatmap indicates the log_2_-(fold change) levels, red: increase; green: decrease. (**C**), The transcription levels of BrERF2 and BrERF109 in growing plants under melatonin treatment. (**D**), Schematic diagram of the effectors, reporters, and internal compositions. (**E**), Transcriptional regulatory activities of BrERF2 and BrERF109. The empty pBD (set as 1) and pBD-VP16 vectors were used as the negative and positive controls, respectively. Each value indicates the mean ± SD of six biological replicates. The asterisks above columns indicate a significant difference at * *p* < 0.05.

**Figure 5 ijms-24-02933-f005:**
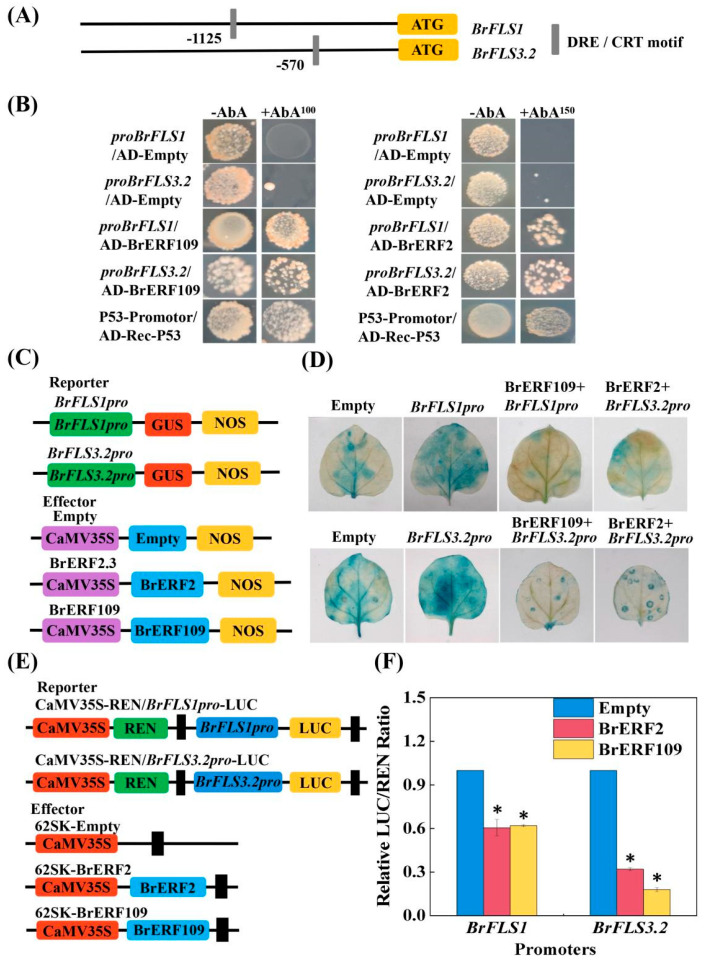
BrERF2 and BrERF109 directly bind to the promoter of *BrFLS1* and *BrFLS3.2*, respectively. (**A**), Promoter schematic diagram of flavonoid biosynthesis genes. (**B**), Y1H analysis indicated that BrERF2 and BrERF109 could directly bind to the promoter of *BrFLS1* and *BrFLS3.2*, respectively. (**C**), Schematic diagram of reporter and effector genes used for GUS staining. (**D**), GUS staining analysis showed that BrERF2/BrERF109 inhibited the expression of *BrFLS1* and *BrFLS3.2,* respectively. (**E**), Schematic diagram of reporter and effector genes used for LUC assay. (**F**), Relative LUC/REN ratio indicated that the transcription levels of *BrFLS1* and *BrFLS3.2* were suppressed by BrERF2/BrERF109. The asterisks above columns indicate a significant difference at * *p* < 0.05. Each value indicates the mean ± SD of six biological replicates.

**Figure 6 ijms-24-02933-f006:**
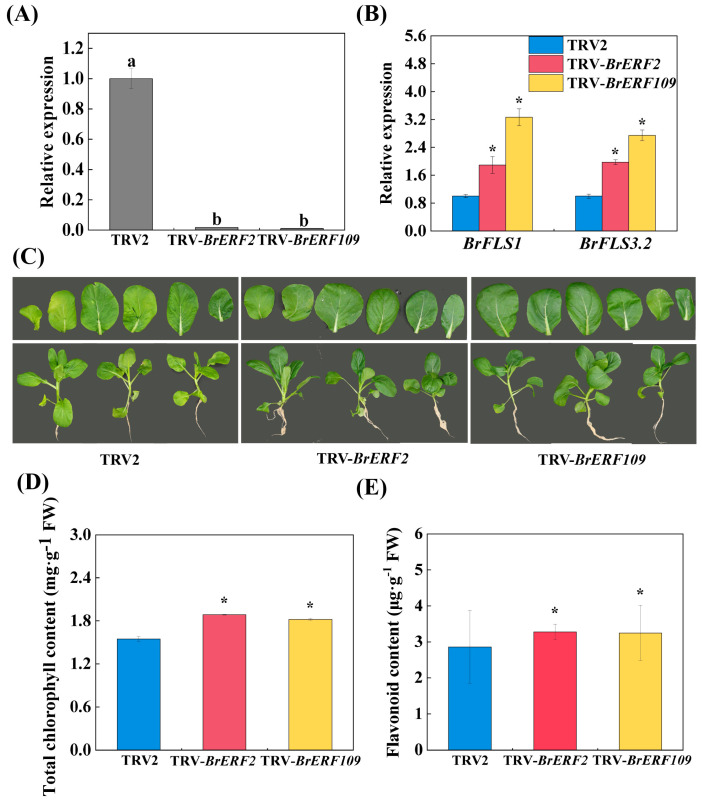
VIGS assay of BrERF2 and BrERF109 in flowering Chinese cabbage. (**A**), The transcription levels of *BrERF2* and *BrERF109* in TRV-*BrERF2* and TRV-*BrERF109* plants. (**B**), The expressions of *BrFLS1* and *BrFLS3.2* in TRV2, TRV-BrERF2 and TRV-BrERF109 plants. (**C**), The plant and leaves phenotypes of TRV2, TRV-*BrERF2* and TRV-*BrERF109* plants. (**D**), Total chlorophyll content of TRV2, TRV-*BrERF2* and TRV-*BrERF109* plants. (E), Flavonoid content of TRV2, TRV-*BrERF2* and TRV-*BrERF109* plants. Each value indicates the mean ± SD of three biological replicates. The asterisks above columns indicate a significant difference at * *p* < 0.05.

**Figure 7 ijms-24-02933-f007:**
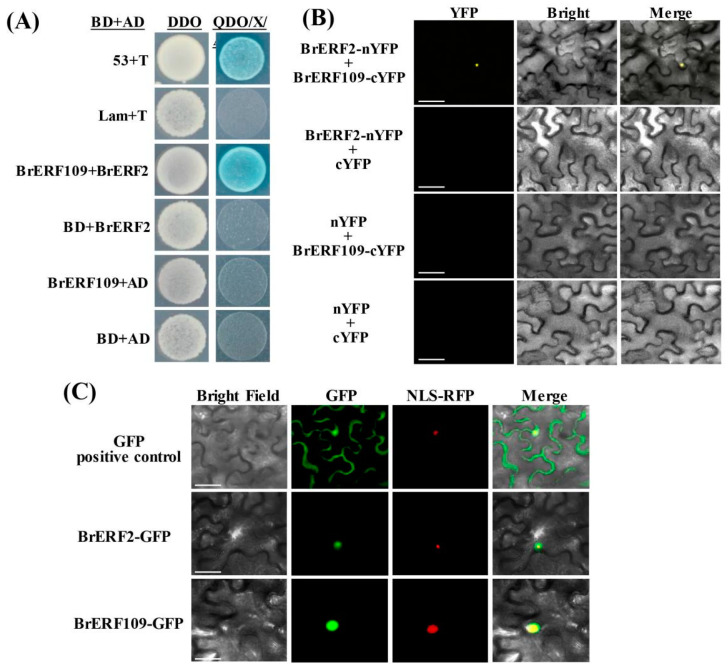
BrERF2 interacts with BrERF109. (**A**), The interaction between BrERF2 and BrERF109 was analyzed by Y2H assay. (**B**), BiFC analysis showed the interaction of BrERF2 and BrERF109 in vivo. Bar, 25 μm. (**C**), Subcellular localization of BrERF2 and BrERF109 in tobacco leaves. Bars, 25 μm. Each protein-protein interaction was proved by three independent experiments.

## Data Availability

Not applicable.

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
