# Peer review of "Melatonin Delays Postharvest Senescence through Suppressing the Inhibition of BrERF2/BrERF109 on Flavonoid Biosynthesis in Flowering Chinese Cabbage"

_ijms, 2023, doi:10.3390/ijms24032933_

Round 1

Reviewer 1 Report

The following suggestions to be addressed by the authors before the acceptance of the manuscript.

(1) Line No. 12: Correct as 'remarkable role'

(2) Line No. 15-16: The sentence is incomplete

(3) Difference in font size among the sentences is observed in the body of the manuscript.

(4) Line No. 60: Rewrite as 'Flavonoid, a class of secondary..'

(5) Line No. 63: Correct as 'The high level..'

(6) Line No. 85-92: This text should be a part of the results and not the introduction.

(7) Line No. 116: Put the reference number inside bracket.

(8) Line No. 333: Cross check the reference numbers and present them correctly. Similar error is observed at multiple places of the manuscript. Kindly correct throughout the manuscript.

(9) Line No. 492: Italicise 'Nicotiana benthamiana'

(10) Line No. 525: Correct the spell error in the sib-heading 4.9.

(11) Line No. 563: Rewrite the statement as 'Details of the primers involved in'

Author Response

  Thank you very much for your responsible and professional comments. We have revised the manuscript according to your suggestions. Our response to your comments has been listed below point-by-point.

Question 1: Line No. 12: Correct as 'remarkable role'

Response: Dear reviewer, we so appreciate your responsible and professional comments on our manuscript. We have revised the word and highlighted it on page 1 lines 12.

Question 2: Line No. 15-16: The sentence is incomplete

Response: Dear reviewer, thanks for your suggestion. We have rewritten the sentence and highlighted it on page 1 lines 14-15. 

Question 3: Difference in font size among the sentences is observed in the body of the manuscript.

Response: Dear reviewer, thanks for your responsible comments. We have examined and revised them throughout the manuscript such as page 2 lines 88-93.

Question 4: Line No. 60: Rewrite as 'Flavonoid, a class of secondary..'

Response: Dear reviewer, thanks for your suggestion. We have rewritten the word and highlighted it on page 2 lines 69.

Question 5: Line No. 63: Correct as 'The high level..'

Response: Dear reviewer, thanks for your suggestion. We have rewritten the word and highlighted it on page 2 lines 72.

Question 6: Line No. 85-92: This text should be a part of the results and not the introduction.

Response: Dear reviewer, thanks for your suggestion. We have deleted this part.

Question 7: Line No. 116: Put the reference number inside bracket.

Response: Dear reviewer, thanks for your responsible and professional comments. We have revised it on page 3 lines 117.

Question 8: Line No. 333: Cross check the reference numbers and present them correctly. Similar error is observed at multiple places of the manuscript. Kindly correct throughout the manuscript.

Response: Dear reviewer, thanks for your responsible and professional comments. We have examined and revised them throughout the manuscript.

Question 9: Line No. 492: Italicise 'Nicotiana benthamiana'

Response: Dear reviewer, thanks for your professional comment. We have revised it on page 15 lines 492.

Question 10: Line No. 525: Correct the spell error in the sib-heading 4.9.

Response: Dear reviewer, thanks for your professional suggestion. We have revised it on page 15 lines 525.

Question 11: Line No. 563: Rewrite the statement as 'Details of the primers involved in'

Response: Dear reviewer, thanks for your professional suggestion. We have revised it on page 16 lines 563.

Reviewer 2 Report

In the current study, Melatonin response through suppressing the inhibition of BrERF2/BrERF109 on flavonoid biosynthesis in flowering Chinese cabbage, indicating that it delays postharvest senescence. In addition, the authors also molecular regulatory network of melatonin delaying leaf senescence and initially ascertained that melatonin promoted flavonoid accumulation by suppressing the inhibition of BrERF2/BrERF109 on the transcripts of BrFLS1 and BrFLS3.2, which led to delaying the leaf senescence of postharvest flowering Chinese cabbage. The overall manuscript is well organized and well-written. However, the main concerns about this manuscript can be found below, and minor revision are suggested.

Check line 14 in the abstract for typo.

In the abstract the authors should briefly describe the methods and techniques used for characterizations.

Add pathway of melatonin production and main genes involve in melatonin production leading to delayed senescence in line 48-line 59. The following articles would be helpful to cite.  https://doi.org/10.3390/genes13101699,

Line 56-59 check typos.

Line 60-61 the sentence is not clear please revise.

Text type should be consistent in the whole MS.

Section 4.5 should be cited with recent study

https://doi.org/10.1016/j.indcrop.2022.116090

conclusion should be elaborated to add future recommendations.

Add significance of the study in the conclusion.

Author Response

  Thank you very much for your responsible and professional comments. We have revised the manuscript according to your suggestions. Our response to your comments has been listed below point-by-point.

Question 1: Check line 14 in the abstract for typo.

Response: Dear reviewer, thanks for your responsible and professional suggestion. We have unified the font and highlighted it on page 1 lines 13.

Question 2: In the abstract the authors should briefly describe the methods and techniques used for characterizations.

Response: Dear reviewer, we so appreciate your responsible and professional comments on our manuscript. We have added and highlighted it on page 1 lines 17-20.

Question 3: Add pathway of melatonin production and main genes involve in melatonin production leading to delayed senescence in line 48-line 59. The following articles would be helpful to cite.  https://doi.org/10.3390/genes13101699

Response: Dear reviewer, thanks for your constructive advice. We have added according to your suggestion, and these corrections are marked in red in the revised paper on page 2 lines 50-58.

Question 4: Line 56-59 check typos.

Response: Dear reviewer, thanks for your professional suggestion. We have unified the font on page 2 lines 66-68.

Question 5: Line 60-61 the sentence is not clear please revise.

Response: Dear reviewer, thanks for your suggestion. We have revised and highlighted it on page 2 lines 69-70.

Question 6: Text type should be consistent in the whole MS.

Response: Dear reviewer, thanks for your professional suggestion. We have unified the font throughout the manuscript.

Question 7: Section 4.5 should be cited with recent study    https://doi.org/10.1016/j.indcrop.2022.116090

Response: Dear reviewer, as you suggested, we have changed it.

Question 8: Conclusion should be elaborated to add future recommendations. Add significance of the study in the conclusion.

Response: Dear reviewer, as you suggested, we have added on page 17 lines 578-581.